# Differences in Lumbar–Pelvic Rhythm Between Sedentary Office Workers with and Without Low Back Pain: A Cross-Sectional Study

**DOI:** 10.3390/healthcare13101135

**Published:** 2025-05-13

**Authors:** Takaaki Nishimura, Masayasu Tanaka, Natsuko Morikoshi, Tamaki Yoshizawa, Ryo Miyachi

**Affiliations:** 1Faculty of Health and Medical Sciences, Hokuriku University, 1-1 Taiyogaoka, Kanazawa 920-1180, Japan; ry_miyachi@hokuriku-u.ac.jp; 2Department of Community-Based Rehabilitation, Nanto Municipal Hospital, 938 Inami, Nanto 932-0211, Japan; 3Nanto City Home-Visit Nursing Station, 938 Inami, Nanto 932-0211, Japan; hayateno0207@gmail.com; 4Department of Pediatrics, Takaoka City Hospital, 4-1 Takaramachi, Takaoka 933-8550, Japan; tatetsuru15@gmail.com; 5Department of Nursing, Nanto Municipal Hospital, 938 Inami, Nanto 932-0211, Japan; tamaki362@gmail.com

**Keywords:** low back pain, sedentary office workers, lumbar–pelvic rhythm

## Abstract

Background/Objectives: Sedentary office workers (SOWs) often adopt prolonged sitting postures, which potentially disrupt the lumbar–pelvic rhythm (LPR) and contribute to lower back pain (LBP). This study aimed to clarify the group differences in LPR and related physical factors between SOWs with and without LBP. Methods: Sixty-three SOWs were divided into LBP (*n* = 30) and non-LBP (*n* = 33) groups. The lumbar flexion angle (LF) and lumbar–hip angle difference (LHD), which are indicators of LPR, were measured using inertial sensors during trunk flexion. Hip flexion muscle strength (HFMS) and hip-extension muscle strength (HEMS) were assessed using handheld dynamometry. Hip joint range of motion (ROM) was measured using a goniometer. Lumbar proprioception was evaluated via active joint repositioning, and pain and perception were assessed using the Visual Analog Scale, Oswestry Disability Index, and Fremantle Back Awareness Questionnaire. Results: Multiple regression analysis showed significantly greater LF (estimated regression coefficient [ERC]: −2.9, *p* < 0.05) and LHD (ERC: −5.5, *p* < 0.05) during early trunk flexion (ETF) in the LBP group. In the LBP group, LHD during ETF and late trunk flexion were positively correlated with HFMS, and HFMS was correlated with HEMS. Conclusions: HFMS may contribute to an altered LPR in SOWs with LBP.

## 1. Introduction

In modern society, working environments centered around deskwork have become commonplace, and many workers spend increasing amounts of time in seated positions [1]. Lower back pain (LBP), one of the most common occupational condition [2], has become increasingly prevalent among sedentary office workers (SOWs) because of habitual trunk flexion and prolonged sitting [3]. LBP among SOWs can result in reduced labor productivity and increased medical costs [4,5,6], making it essential to identify the contributing factors and develop effective preventive strategies.

In this study, we focused on nonspecific LBP (NSLBP), which refers to LBP not attributed to a specific underlying pathology, such as fracture, infection, or malignancy. NSLBP accounts for the majority of LBP cases and is particularly relevant in occupational populations such as SOWs, where mechanical stress due to prolonged posture plays a substantial role.

Subjective assessment tools, such as the Visual Analog Scale (VAS), Oswestry Disability Index (ODI), and Fremantle Back Awareness Questionnaire (FreBAQ), have been widely used to evaluate pain intensity, functional disability, and altered body perception in individuals with LBP across various occupational groups [7,8,9]. For instance, VAS and ODI have been applied to healthcare workers, manual laborers, and manufacturing workers to assess the musculoskeletal burden resulting from repetitive strain or awkward postures [7,8]. The FreBAQ has been used to assess disturbed body schemas in workers with chronic LBP, including nurses and caregivers [9]. These findings suggest that occupational context influences both the physical and psychological dimensions of LBP.

Although these subjective measures are commonly used, the assessment of LBP in this population predominantly relies on self-reported outcomes. Objective biomechanical assessments, such as those evaluating movement coordination, joint flexibility, and muscle strength, remain limited in SOWs. Given that prolonged sitting can lead to lumbar muscle atrophy [3], reduced hip flexibility [10], decreased hip muscle strength [11], impaired proprioceptive acuity in the lumbar region, as measured by active joint repositioning sense (AJRS) [12], as well as increased reliance on passive stabilizing structures [13,14], it is essential to incorporate such objective evaluations alongside subjective tools to obtain a more comprehensive understanding of LBP in this occupational group.

Among the various biomechanical factors involved in LBP, an abnormality in the lumbar-pelvic rhythm (LPR) is considered a key contributor [15]. LPR refers to a coordinated movement pattern of the lumbar spine, pelvis, and hip joints during trunk motion [16]. When a substantial imbalance exists in the relative flexibility or mobility of these segments, mechanical stress tends to concentrate in the hypermobile region, thereby increasing the risk of tissue overload and pain [17]. Therefore, assessing LPR in individuals with LBP is crucial for understanding the underlying movement dysfunctions and for planning effective interventions.

Previous studies have compared LPR between individuals with and without LBP and reported significant alterations in the timing and contribution of spinal and pelvic movements in patients with LBP [16,18]. However, these studies have largely focused on the general population or physically active workers, and evidence specific to SOWs remains scarce, despite the increasing prevalence of LBP in sedentary occupations.

Although the main risk factor for LBP in SOWs is prolonged sitting, this static posture can lead to secondary biomechanical adaptations that manifest during dynamic movements, such as trunk flexion and extension. Therefore, the present study focused on movement tasks to investigate how such adaptations affect LPR. Understanding the movement characteristics of SOWs may help bridge the gap between static occupational exposure and dynamic motor dysfunction associated with LBP.

Owing to the nature of their work, SOWs tend to adopt a sustained trunk flexion posture during prolonged sitting [19]. This posture has been associated with erector spinae muscle atrophy [3] and increased reliance on passive support structures, such as ligaments, rather than active muscular control [13,14]. Additionally, prolonged sitting reduces the flexibility of the surrounding tissues at the hip joint, including the iliopsoas and hamstrings [10,20], which may contribute to excessive compensatory lumbar motion during dynamic tasks, such as trunk flexion [21]. These biomechanical adaptations may have played a role in the development and persistence of LBP in this population. However, the differences in LPR during movement between SOWs with and without LBP have not yet been clarified. Moreover, physical and perceptual factors associated with LPR, such as muscle strength, hip joint range of motion, and proprioceptive function, have not been sufficiently investigated.

To address these issues, a comprehensive evaluation of the mechanical and sensory-motor components of LPR is needed. In the present study, we selected a set of outcome measures based on prior literature and theoretical rationale to capture the multifactorial aspects of LPR. Hip flexion range of motion (HFROM) and hip-extension range of motion (HEROM) were included because of their influence on pelvic movement and lumbar compensation [22,23]. Muscle strength, including hip flexion muscle strength (HFMS) and hip-extension muscle strength (HEMS), was measured to evaluate the capacity of active muscular support in pelvic–lumbar coordination [24]. Proprioceptive function was assessed using the absolute error in active joint repositioning sense (AEAJRS), which reflects sensory-motor control of lumbar positioning [12]. Additionally, altered body perception was evaluated using the Fremantle Back Awareness Questionnaire (FreBAQ-J), which captures perceptual disturbances commonly associated with chronic LBP [25]. These parameters were selected to clarify not only LPR differences but also the underlying physical and perceptual factors contributing to movement dysfunction in SOWs with LBP.

Therefore, the present study aimed to clarify the differences in LPR between SOWs with and without LBP and to examine associated physical and perceptual factors, including flexibility, muscle strength, and proprioception.

## 2. Materials and Methods

### 2.1. Study Design and Participants

This study used a cross-sectional observational design. The measurements were conducted indoors at gymnasiums and local health facilities. The participants were SOWs (those whose occupational activities involved sitting for 80% or more of their workdays) aged 20–60 years with or without LBP [26]. The exclusion criteria for those with LBP were as follows: (i) LBP due to organic factors, (ii) LBP that occurred within the last three months (acute back pain), (iii) a prior surgical procedure involving the lumbar spine or hip, (iv) severe spinal deformities such as kyphosis, and (v) pregnancy. Participants were recruited via posters and email invitations distributed in local office settings. Interested individuals contacted the research team and were screened for eligibility using a standardized questionnaire. Selection was performed by the research team based on predefined inclusion and exclusion criteria. A prior medical diagnosis of LBP was not required. Participants were classified into groups with or without LBP based on their self-reports at the time of study registration. This group status was treated as the independent variable in the analysis. Participants in the non-LBP group reported no current symptoms of LBP and scored 0 on the Visual Analog Scale (VAS) [27], Oswestry Disability Index (ODI) [28], and FreBAQ-J [25], confirming their classification as asymptomatic.

In accordance with ethical standards, participants received both verbal and written explanations regarding the study’s purpose, procedures, and handling of their personal data. Written informed consent was obtained prior to the commencement of the study.

The required sample size was estimated based on the commonly used guidelines of at least 15 participants per covariate in a multiple regression analysis. Because the model included age, sex, and Body Mass Index (BMI) as covariates, the minimum required sample size was 45. The final sample of 63 participants met this criterion.

### 2.2. Measurement

The evaluation items were classified into the following three categories: (1) subjective assessments, including the level of LBP, as measured by the VAS [27]; abnormal physical perception of the lumbar region, as assessed by the FreBAQ-J [25]; and the impact of LBP on daily life, as assessed by the ODI [28]; (2) physical function assessments related to joint range of motion, including lumbar flexion angle (LF), lumbar–hip angle difference (LHD), HFROM, HEROM [22,23], and lumbar proprioception, as assessed using the AEAJRS [12]; and (3) muscle strength, including HFMS and HEMS [24,29].

The VAS is a scale used to evaluate the subjective intensity of pain and indicates the degree of one’s own pain on a straight line from 0 to 100 mm [27]. This scale is simple to use, highly sensitive, and can capture detailed changes in pain. It has been confirmed to be reliable and valid for evaluating LBP, and it is widely used in clinical research and medical practice settings [30].

The FreBAQ-J is a 9-item questionnaire developed by Nishigami et al. [25] to assess abnormal physical perception in the lumbar region. A higher total FreBAQ-J score indicated greater abnormal physical perception in the lumbar region. In this study, we calculated total scores.

The ODI consists of 10 items, each of which is scored on a scale of 0 to 5, and the ODI score (percentage) increases as the impact of LBP on daily life increases [28]. The ODI was calculated as a percentage of the maximum score of 50. If there were unanswered items, the item was excluded and the percentage was calculated from the total score of the remaining items.

Lumbar and hip flexion angles were measured using inertial sensors (TSND151, ATR-Promotions, Sagara, Japan) in combination with data acquisition software (Sensor Controller version 3.0.0, ATR-Promotions, Sagara, Japan). The inertial motion sensors were placed at the following three locations: the thoracolumbar transition (above the first lumbar vertebra), lumbar transition (above the first sacral vertebra), and right thigh (posterior surface of the thigh between the ischial tuberosity and popliteal fossa) (Figure 1a). The inertial motion sensors were set to an acceleration range of ±8 G, angular velocity range of ±1000 dps, and sampling interval of 10 ms. After standing still for 5 s, the participants performed a maximum trunk flexion in 3 s, and held the final position for 3 s (Figure 1b). The participants were given a full explanation of the procedure and practiced the movements before the measurement. Regarding the reproducibility of the measurement, Nishimura et al. [29] reported an ICC of 0.8 or higher for all forward bending angles. The reproducibility was ensured in the present study. Trunk flexion angle was defined as the inclination angle of the sensor in the thoracolumbar transition area. The lumbar and hip flexion angles were recorded as 30° in early trunk flexion (ETF) and 60° in late trunk flexion (LTF). Lumbar flexion angle was defined as the difference between the inclination angles of the sensors at the thoracolumbar and lumbosacral transitions at each trunk flexion angle. The hip flexion angle was defined as the difference between the inclination angles of the sensors at the lumbosacral transition and thigh sensors. The values obtained were defined as positive and negative for flexion and extension, respectively. The difference between the flexion angles of the lumbar spine and hip joint was calculated as an indicator of LPR (lumbar–hip angle difference [LHD]: lumbar flexion angle—hip flexion angle). The average of three measurements was considered as the representative value for each participant.

The hip joint range of motion (ROM) was measured using a goniometer (plastic angle meter; ÖSSUR Japan G.K., Tokyo, Japan). The HFROM was measured in the supine position with the basic axis parallel to the trunk and the moving axis connecting the greater trochanter and lateral epicondyle of the femur. The knee joint was in maximum flexion (Figure 2a). HEROM was measured in the prone position, with the basic and moving axes the same as those for the hip flexion measurement and the knee joint in the full extension position (Figure 2b). Each measurement was performed once.

Lumbar proprioception was assessed using the active joint repositioning sense (AJRS) method during trunk flexion in a standing position [12]. Inertial motion sensors were used in the same manner as for the lumbar and hip joint angle measurements. The starting position was standing with the eyes closed and arms crossed in front of the chest (Figure 3a). Participants were first shown a target position of 20° trunk flexion and asked to memorize it. Subsequently, participants were instructed to reproduce a 20° lumbar flexion angle with their eyes closed and without visual cues (Figure 3b).

The reproduced lumbar flexion angle was defined as the average value over a 1 s interval during the final reproduced posture, excluding the first and last seconds. This task was repeated three times. The absolute error (AE) was calculated as the mean of the absolute differences between the reproduced angles and the target angle of 20° in accordance with Brindle et al. [31]. The following formula was used:AE = ∑∣Xi − X true∣/n
where Xi is the reproduced lumbar flexion angle for each trial, X is the target angle (20°), and n is the number of trials. Thus, AE represents the average deviation from the target angle, regardless of the direction, and it serves as an index of lumbar proprioceptive accuracy.

Given that HFMS and HEMS are factors that can influence abnormalities in the LPR [24], both HFMS and HEMS were measured using a handheld dynamometer (HHD, Tas F-1, ANIMA, Chofu, Japan) [29]. Measurements were performed on the dominant leg, which was defined as the leg used to kick the ball. Hip flexion strength was measured with the participants in a sitting position, with both lower limbs raised off the floor, the hip and knee joints bent at 90°, and the HHD placed in front of the thigh, 10 cm proximal to the upper edge of the patella (Figure 4a). Hip-extension strength was measured in the prone position with one knee joint bent at 90° and the other knee joint extended to 0°. The HHD was placed on the posterior aspect of the thigh, 10 cm proximal to the midpoint of the popliteal fossa in the lower limb, and in the 90° knee flexion position (Figure 4b). To ensure consistency in the measurements, the HHD was secured using a belt (Traction Belt 270 cm; Erler Zimmer GmbH & Co. KG, Deutschland, Germany), which was wrapped around the measurement bed and attached to the distal thigh of each participant. During the muscle strength measurements, the subjects were verbally instructed not to make any compensatory movements such as extending the trunk. The measurement procedure was as follows: the subject performed one 5 s maximum isometric contraction, rested for 30 s, and then performed another 5 s maximum isometric contraction. Torque (Nm) was calculated by multiplying the obtained muscle strength data (N) by the femur length (distance from the right greater trochanter to the lateral epicondyle of the right femur), and the torque-to-body weight ratio (Nm/kg) was calculated by normalization to body weight. The maximum value of the two trials was used as the representative muscle strength value during hip flexion and extension.

### 2.3. Statistical Analyses

The statistical analyses were performed using an EZR Version 1.67 [32] and SPSS version 28 (IBM SPSS Statistics, IBM Japan, Tokyo, Japan). For between-group comparisons of the participant characteristics with or without LBP, an unpaired *t*-test was applied to age, BMI, LF and LHD at ETF and LTF, HFROM, HEROM, HFMS, HEMS, and AEAJRS. A chi-squared test was used for sex. Multiple linear regression models were independently constructed to compare each outcome variable between the LBP and no-LBP groups, while adjusting for age, sex, and BMI [33]. Each model included one of the following as the dependent variable: LF, LHD, HFROM, HEROM, HFMS, HEMS, or AEAJRS. Group status (with or without LBP) was included as the main independent variable. For example, when analyzing LF as the dependent variable, the model included group status (with or without LBP), age, sex, and BMI as independent variables. The estimated regression coefficients were used to assess adjusted differences between groups. The normality of residuals was confirmed using Q–Q plots for each dependent variable prior to conducting the regression analyses. This analytical strategy is consistent with standard practice in cross-sectional observational studies and follows the method used in a previous study involving rural-dwelling older adults, in which group status (e.g., with or without exercise habits) was treated as an independent variable to evaluate adjusted group differences [34]. Additionally, partial correlation analyses were conducted separately within the LBP and no-LBP groups to examine associations among continuous outcome variables, controlling for age, sex, and BMI. The significance level for all statistical analyses was set at 0.05.

## 3. Results

The basic characteristics of the participants and measurement results are presented in Table 1. There were 63 participants, including 30 with LBP (10 men and 20 women) and 33 without LBP (14 men and 19 women). No significant differences in age or sex were observed between the groups; however, the BMI was significantly higher in the LBP group than in the non-LBP group. In individuals with LBP, the VAS was 26.1 ± 13.5 mm (maximum: 100 mm), ODI was 15.7 ± 8.7% (maximum: 50 points), and FreBAQ-J was 5.9 ± 5.6 points (maximum: 36 points). Based on previous studies, the VAS score is consistent with mild pain, and the ODI score falls within a range generally interpreted as minimal to moderate disability [35,36]. Regarding the FreBAQ-J, as there are no established cutoff values for severity classification, the score was reported descriptively to reflect altered body perception in the lumbar region [37].

Table 2 shows a comparison of the lumbar flexion angles during ETF and LTF, LHD, HFMS and HEMS, HFROM and HEROM, and AEAJRS according to the presence or absence of LBP. Compared to the non-LBP group, the LBP group demonstrated significantly lower values in hip-extension range of motion (HEROM: 16.6 ± 4.1° vs. 19.1 ± 3.8°), hip flexion muscle strength (HFMS: 1.0 ± 0.3 Nm/kg vs. 1.8 ± 1.9 Nm/kg), and hip-extension muscle strength (HEMS: 0.7 ± 0.3 Nm/kg vs. 1.2 ± 0.4 Nm/kg). No significant differences were observed in the other variables.

Table 3 shows a comparison of the LF during ETF and LTF; LHD, HFMS and HEMS; HFROM and HEROM; and AEAJRS according to the presence or absence of LBP. The multiple regression analysis showed that the lumbar flexion angle and LHD during ETF were significantly greater in the LBP group than in the non-LBP group (estimated regression coefficient [ERC] for lumbar flexion angle: −2.9, *p* < 0.05; ERC for LHD: −5.5, *p* < 0.05). An ERC of −2.9 indicates that, after adjusting for covariates, the LBP group exhibited 2.9° greater lumbar flexion during ETF compared to the non-LBP group. The R-squared value of the regression model for LF in the ETF was 0.14. No significant differences in trunk flexion were found between the groups for the other lumbar flexions. The HFMS and HEMS scores were significantly lower in individuals with LBP than in those without LBP (regression coefficient estimate: 0.4, *p* < 0.05 for HFMS; regression coefficient estimate: 0.4, *p* < 0.05 for HEMS). Additionally, a comparison of the hip joint ROM revealed that only HEROM was significantly smaller in the LBP group than in the non-LBP group (ERC: 2.4, *p* < 0.05).

The results of the partial correlation analysis for the LBP and non-LBP groups are presented in Table 4 and Table 5, respectively. In the LBP group, the LHDs during the ETF and LTF were significantly positively correlated with HFMS (ETF, *r* = 0.42; LTF, *r* = 0.41), and the HFMS was significantly and positively correlated with HEMS (*r* = 0.70). In the non-LBP group, no correlation was noted between LHD and HFMS, or HEMS, or between HFMS and HEMS.

## 4. Discussion

In this study, we examined the differences in LPR between SOWs with and without LBP and the factors related to LPR in patients with LBP.

### 4.1. Differences in LPR Between Individuals with and Without LBP in SOWs

The results of this study show that the lumbar flexion angle and LHD during ETF were significantly greater in participants with LBP than in those without LBP, suggesting that the LBP group may have experienced excessive lumbar flexion during the early phase of movement. Similarly, previous studies report that individuals with LBP exhibit increased lumbar motion during ETF [38,39], which is consistent with the present findings.

The multifidus plays a key role in controlling lumbar spine movements, particularly during the initial stages of trunk motion [40]. Sustained sitting postures have been shown to contribute to the atrophy of the lumbar multifidus muscle [3], and individuals with LBP often demonstrate reduced muscle [41,42]. In particular, delayed activation of the multifidus muscle at the onset of movement has been observed in individuals with LBP [43]. Although the current study did not directly measure the electromyographic (EMG) activity of the multifidus muscle, the observed increase in lumbar flexion may reflect reduced activity or delayed activation of this muscle. However, as no EMG recordings were obtained in this study, this interpretation remains speculative because of the absence of direct EMG data. Caution is warranted when inferring the underlying muscle function, and further studies incorporating EMG evaluation of the multifidus muscle are needed to clarify its contribution to LPR abnormalities.

Additionally, individuals with LBP have frequently been reported to exhibit hamstring shortening, which can inhibit anterior pelvic tilt during trunk flexion, compared with those without LBP [44,45]. A prolonged sitting time has also been associated with increased hamstring stiffness [20]. Thus, in addition to multifidus muscle dysfunction, reduced hamstring flexibility in SOWs with LBP may have contributed to limited anterior pelvic rotation (hip flexion), leading to compensatory increases in lumbar flexion during ETF. However, as hamstring flexibility was not directly assessed in this study, this explanation remains hypothetical because of the lack of direct measurement of hamstring flexibility. Future research is warranted to examine hamstring flexibility in relation to lumbar–pelvic movement patterns in this population.

Although the present findings suggest biomechanical adaptations involving the lumbar spine and hip in SOWs with LBP, the lack of direct assessment of muscle activity and flexibility limits the strength of these interpretations. Further studies incorporating objective evaluations such as EMG and range-of-motion testing are necessary to confirm the underlying mechanisms.

### 4.2. Differences in Factors Related to LPR in Individuals with and Without LBP

The results indicate associations between HFMS and LHD during ETF and LTF, as well as between HFMS and HEMS, in individuals with LBP. In contrast, no associations between LHD and HFMS or HEMS, or between HFMS and HEMS, were found in individuals without LBP. Individuals with LBP are more likely to lack stability in the hip and pelvic girdle, such as in the activity of the hip flexion and extension muscles and fixation of the sacroiliac joint [46,47]. Hip flexion and extension muscle activities contribute to pelvic stability through femoral control [48,49]. From these results, only the LBP group showed a positive correlation between HFMS and HEMS; therefore, SOWs with LBP may compensate for excessive lumbar flexion through increased engagement of hip flexors and extensors to stabilize the pelvis and facilitate trunk flexion. However, further investigation is needed to determine the characteristics of the LPR in individuals with LBP who have reduced hip muscle activity and strength during movement, as well as impaired lumbar stability.

No significant association was observed between the LHD and lumbar proprioception. Lumbar motor control adjusts muscle output based on sensory information [50], of which proprioception, which detects the position and movement of the body, is particularly important [51]. Several factors may explain the lack of correlation observed in this study. First, the task used to assess LHD involved instructed trunk flexion without the need for precise or conscious control of lumbar movements [52]. Therefore, it is possible that the participants did not actively engage in proprioceptive feedback mechanisms during the task, which may have reduced the sensitivity of detecting associations. Second, LHD reflects a kinematic outcome resulting from a combination of multiple biomechanical factors, including joint mobility, segmental coordination, and muscular control, all of which are directly regulated by proprioception [53]. Third, the accuracy of the proprioceptive measures used (active joint repositioning) may depend on the type of movement and cognitive engagement required, which may not have been fully elicited in the task design [31]. Finally, although proprioception is often reduced in individuals with LBP [54], its effect on dynamic coordination during functional trunk motion may be subtle and context dependent.

Taken together, these considerations suggest that future studies should incorporate proprioception tasks that require active control of lumbar motion and assess their relationship with dynamic trunk coordination measures such as LHD under various motor conditions.

### 4.3. Clinical Significance

The results of this study suggest that SOWs with LBP have an abnormality in LPR, with the lumbar spine excessively flexed during ETF, and that HFMS is involved in the abnormality of the LPR in the LBP group. In SOWs with LBP, the ability to control the lumbar spine during ETF is low. Therefore, training that emphasizes isolated hip joint movement while minimizing lumbar spine motion in the early phase may be useful. These findings may inform the development of targeted interventions for SOWs with LBP.

From clinical and ergonomic perspectives, these findings suggest that interventions for SOWs with LBP should focus on reducing excessive lumbar flexion during the initial phase of trunk movement. Physical therapy programs emphasizing hip mobility and strength, particularly targeting the hip flexors and extensors, may help to restore proper lumbar–pelvic coordination. Additionally, ergonomic modifications, such as the use of adjustable sit–stand desks, lumbar support cushions, or posture reminders, may reduce prolonged trunk flexion while sitting, thereby minimizing stress on the lumbar spine. A combination of movement retraining and ergonomic interventions could prevent LBP recurrence or progression in this occupational group.

### 4.4. Study Limitations

In this study, we used LHD as an indicator of LPR, but we did not evaluate the muscle activity of the lumbar multifidus muscle or hip flexion/extension muscles (such as the iliopsoas and gluteus maximus) that are involved in segmental lumbar motion and hip flexion/extension motion, which affect LHD; therefore, further verification is needed. In addition, although LPR and sitting posture are related to each other [55], this study did not examine the effects of sitting posture while working. Therefore, future studies should examine the relationship between sitting posture and LPRs.

## 5. Conclusions

This study investigated the differences in LPR between SOWs with and without LBP and examined potential contributing factors to LPR abnormalities in individuals with LBP. The results show that the lumbar flexion angle and LHD during ETF were significantly greater in the LBP group than in the non-LBP group. Furthermore, the involvement of HFMS in the abnormality of the LPR in the LBP group was suggested. These findings may inform the development of targeted physical therapy interventions and ergonomic strategies to improve lumbar–pelvic coordination and prevent LBP in sedentary occupational populations.

## Figures and Tables

**Figure 1 healthcare-13-01135-f001:**
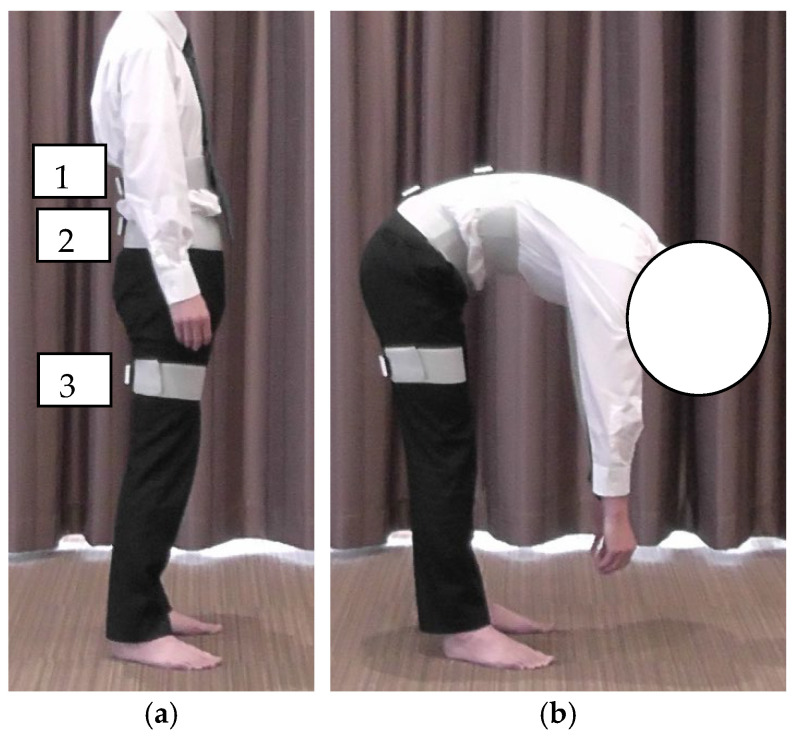
Method of trunk flexion measurement: (**a**) starting position; (**b**) maximum trunk flexion position. (**a-1**) The top edge of the sensor was placed at the top edge of the first lumbar vertebra; (**a-2**) the top edge of the sensor was placed at the top edge of the first sacral vertebra; (**a-3**) midpoint of the sciatic tuberosity and the popliteal fossa.

**Figure 2 healthcare-13-01135-f002:**
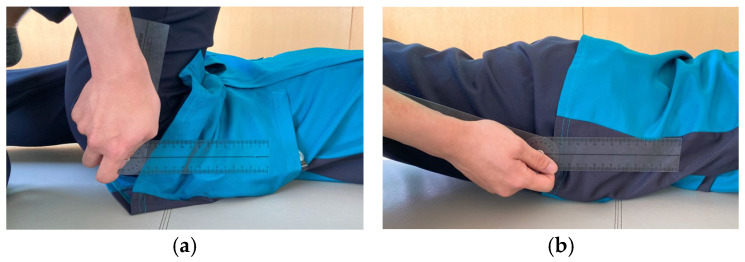
Method for measuring hip flexion (**a**) and extension range of motion (**b**).

**Figure 3 healthcare-13-01135-f003:**
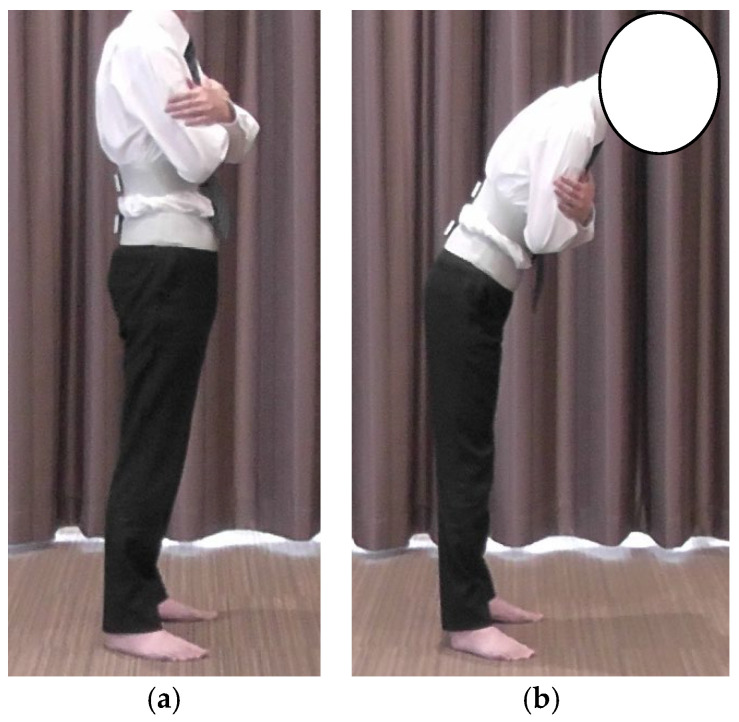
Method of AJRS measurement: (**a**) starting position; (**b**) 20° trunk flexion position. AJRS, active joint repositioning sense.

**Figure 4 healthcare-13-01135-f004:**
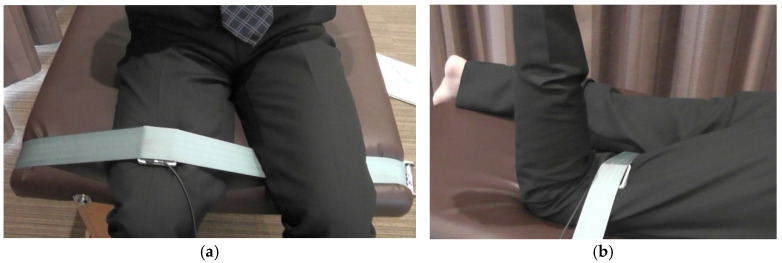
How to measure HFMS and HEMS: (**a**) HFMS—the HHD was placed in front of the thigh, 10 cm proximal to the upper edge of the patella, with the hip and knee joints bent at 90°; the measurement was taken with the HHD secured in place with a belt; (**b**) HEMS—the HHD was placed on the posterior aspect of the thigh, 10 cm proximal to the center of the popliteal fossa, in the 90° flexed knee position. HEMS, hip-extension muscle strength; HFMS, hip flexion muscle strength; HHD, handheld dynamometer.

**Table 1 healthcare-13-01135-t001:** Characteristics of the participants.

ITEMS	LBP Group (*n* = 30)	Non-LBP Group (*n* = 33)	*p*-Value
Age (years)	45.6. (8.7)	41.8 (10.4)	0.12
Sex, *n* (%)	Male, 10 (33.3)	Male, 14 (42.4)	0.60
	Female, 20 (66.7)	Female, 20 (66.7)	
BMI (kg/m^2^)	24.1 (3.3)	21.5 (2.5)	<0.05
VAS for LBP (mm)	26.1 (13.5)	0	<0.05
ODI (%)	15.7 (8.7)	0	<0.05
FreBAQ (total score)	5.9 (5.6)	0	<0.05

Values are presented as means (standard deviations) or numbers (percentages). BMI, Body Mass Index; VAS, Visual Analog Scale; LBP, low back pain; ODI, Oswestry Disability Index; FreBAQ, Fremantle Back Awareness Questionnaire.

**Table 2 healthcare-13-01135-t002:** Comparison of the lumbar flexion angle, LHD during trunk flexion, hip flexion/extension strength, hip flexion/extension range of motion, and absolute error of active joint repositioning sense between with and without LBP.

ITEM	LBP Group (*n* = 30)	Non-LBP Group (*n* = 33)	*p*-Value
LF at ETF (°)	12.7 (4.7)	10.9 (4.3)	0.13
LF at LTF (°)	27.2 (9.2)	26.6 (8.7)	0.80
LHD in ETF (°)	−7.8 (8.6)	−10.9 (9.0)	0.16
LHD in LTF (°)	−14.3 (18.8)	−15.4 (17.7)	0.80
HFROM (°)	108.1 (7.4)	110.0 (7.3)	0.34
HEROM (°)	16.6 (4.1)	19.1 (3.8)	<0.05
HFMS (Nm/kg)	1.0 (0.3)	1.8 (1.9)	<0.05
HEMS (Nm/kg)	0.7 (0.3)	1.2 (0.4)	<0.05
AEAJRS (°)	6.3 (4.9)	7.6 (5.2)	0.32

Values are presented as means (standard deviations). LF, lumbar flexion angle; ETF, early trunk flexion; LTF, late trunk flexion; LHD, lumbar–hip angle difference; HEROM, hip-extension range of motion; HFROM, hip flexion range of motion; HFMS, hip flexion muscle strength; HEMS, hip-extension muscle strength; AEAJRS, absolute error of active joint repositioning sense.

**Table 3 healthcare-13-01135-t003:** Comparison of lumbar flexion angles, LHD during trunk flexion, hip flexion/extension strength, hip flexion/extension range of motion, and absolute error of active joint repositioning sense between those with and without LBP, adjusted for age, BMI, and sex.

ITEMS	ERC	95% CI Lower	95% CI Upper	*p*-Value
LF at ETF (°)	−2.9	−5.3	−0.4	<0.05
LF at LTF (°)	−0.2	−0.4	0.1	0.14
LHD in ETF (°)	−5.5	−10.3	−0.6	<0.05
LHD in LTF (°)	−7.0	−16.6	2.6	0.15
HFROM (°)	1.1	−2.2	4.4	0.5
HEROM (°)	2.4	0.3	4.6	<0.05
HFMS (Nm/kg)	0.4	0.2	0.7	<0.05
HEMS (Nm/kg)	0.4	0.3	0.8	<0.05
AEAJRS (°)	0.3	−0.2	0.7	0.21

Adjusted for age, BMI, and sex. LF, lumbar flexion angle; ETF, early trunk flexion; LTF, late trunk flexion; LHD, lumbar–hip angle difference; HEROM, hip-extension range of motion; HFROM, hip flexion range of motion; HFMS, hip flexion muscle strength; HEMS, hip-extension muscle strength; AEAJRS, absolute error of active joint repositioning sense; ERC, estimated regression coefficient; CI, confidence interval. Regression coefficient estimates for LF in LTF, HFMS, HEMS, and AEAJRS were calculated after logarithmic transformation.

**Table 4 healthcare-13-01135-t004:** Factors related to lumbar–pelvic rhythm in individuals with LBP.

	ODI (%)	FreBAQ (point)	HFMS (Nm/kg)	HEMS (Nm/kg)	HEROM (°)	HFROM (°)	LF at ETF (°)	LF at LTF (°)	LHD in ETF	LHD in LTF	AEAJRS
VAS (mm)	0.62 *	0.47 *	−0.08	−0.01	0.02	0.20	0.14	0.07	0.04	−0.01	0.30
ODI (%)		0.67 *	−0.11	−0.06	−0.07	0.16	−0.08	−0.15	−0.08	−0.20	0.27
FreBAQ (point)			−0.07	−0.07	−0.02	−0.19	0.15	0.05	0.19	0.03	0.15
HFMS (Nm/kg)				0.70 *	0.30	0.31	0.37	0.35	0.42 *	0.41 *	−0.05
HEMS (Nm/kg)					0.43 *	0.22	0.09	0.12	0.18	0.21	0.06
HEROM (°)						0.02	−0.10	−0.01	−0.05	0.05	−0.19
HFROM (°)							−0.21	−0.23	−0.31	−0.29	−0.07
LF at ETF (°)								0.95 *	0.94 *	0.94 *	0.25
LF at LTF (°)									0.87 *	0.98 *	0.20
LHD in ETF										0.89 *	0.22
LHD in LTF											0.15

* Significant correlations. Adjusted for age, BMI, and sex. VAS, Visual Analog Scale; ODI, Oswestry Disability Index; FreBAQ, Fremantle Back Awareness Questionnaire; HFMS, hip flexion muscle strength; HEMS, hip-extension muscle strength; HEROM, hip-extension range of motion; HFROM, hip flexion range of motion; LF, lumbar flexion angle; ETF, early trunk flexion; LTF, late trunk flexion; LHD, lumbar hip angle difference; AEAJRS, absolute error of the active joint repositioning sense.

**Table 5 healthcare-13-01135-t005:** Factors related to lumbar–pelvic rhythm in individuals without LBP.

	HEMS (Nm/kg)	HEROM (°)	HFROM (°)	LF at ETF (°)	LF at LTF (°)	LHD in ETF	LHD in LTF	AEAJRS
HFMS (Nm/kg)	0.04	−0.11	0.14	0.06	0.01	0.06	0.02	0.05
HEMS (Nm/kg)		−0.19	−0.03	−0.07	−0.07	0.03	−0.01	−0.01
HEROM (°)			0.24	−0.07	−0.15	−0.11	−0.9	−0.32
HFROM (°)				−0.25	−0.41 *	−0.30	−0.41 *	−0.13
LF at ETF (°)					0.96 *	0.95 *	0.94 *	0.05
LF at LTF (°)						0.92 *	0.98 *	0.10
LHD in ETF							0.96 *	−0.03
LHD in LTF								0.07

* Significant correlations. Adjusted for age, BMI, and sex. HFMS, hip flexion muscle strength; HEMS, hip-extension muscle strength; HEROM, hip-extension range of motion; HFROM, hip flexion range of motion; LF, lumbar flexion angle; ETF, early trunk flexion; LTF, late trunk flexion; LHD, lumbar–hip angle difference; AEAJRS, absolute error of active joint repositioning sense.

## Data Availability

The data presented in this study are openly available in Zenodo at https://doi.org/10.5281/zenodo.15099268 (accessed on 28 March 2025).

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
