# Peer review of "Differences in Lumbar–Pelvic Rhythm Between Sedentary Office Workers with and Without Low Back Pain: A Cross-Sectional Study"

_healthcare, 2025, doi:10.3390/healthcare13101135_

Round 1
Reviewer 1 Report
Comments and Suggestions for Authors
1. Introduction
(1) It is recommended to discuss the literature on VAS, ODI, and FreBAQ for workers in different occupations.
(2) Please explain why this study measured HFMS, HEMS, HFROM, HEROM, and AJRS.
2.2. Measurement
It is recommended to describe the three themes in order: subjective rating (VAS, ODI, FreBAQ-J), joint range of motion (LF, LHD, HFROM, HEROM, AEAJRS), and muscle strength (HFMS, HEMS).
2.3. Statistical Analyses
(1) “Regression coefficient estimates in the multiple regression analysis were then calculated to compare the differences between the groups.” A regression coefficient in multiple regression is the slope of the linear relationship between the criterion variable and the part of a predictor variable that is independent of all other predictor variables. Therefore, to me, regression coefficients are not a suitable indicator for comparing different groups.
(2) In regression analysis, which variables are independent variables and which variable is the dependent variable?
(3) Why not use independent T test to test the difference between the groups with and without LBP?
3. Results
(1) Please present the means and standard deviations for all measurement items.
(2) The VAS for LBP of the low back pain group is 26.1 mm (maximum 100 mm), ODI is 15.7% (maximum 50 point), and FreBAQ is 5.9 point (maximum 36 point). Does these results indicate that the pain is severe? What are the results of the Non-LBP group in these three indicators?
(3) Table 2, the ERC value of LF at ETF is "-2.9". What does this value mean? What is the R square value of the regression model?
Reviewer 2 Report
Comments and Suggestions for Authors
Reviewer Comments:
I appreciate the opportunity to review the manuscript entitled “Differences in Lumbar-Pelvic Rhythm Among Sedentary Office Workers With and Without Low Back Pain”. The study addresses a topic of high clinical relevance, especially in the current context of a sedentary work environment.
1. Title:
- Detail the type of study
2. Abstract
- indicate some statistical results. In addition, the difference between groups both in methodology and in results
3. Introduction
- Well grounded in previous literature, but somewhat repetitive in terms of the relevance of LBP in sedentary workers. Consolidate paragraphs to avoid redundancy.
- It would be good to clarify the type of low back pain. To make a contextualization about it.
4. Methods
- Detail how participants accessed the study. Who made the selection and if they had a previous diagnosis.
-It is recommended to briefly describe how the sample size was calculated, if done.
-How did you obtain the value of the result of the error of the lumbar resposioning? the difference between the attempts? average? the result was the 20º by subtracting the participant's attempt?
5. Dicussion and conclusion
- Although the role of the multifidus muscle is discussed, its electromyographic (EMG) activity is not evaluated, which limits mechanical interpretation. how can they know if multifidus muscle activity is decreased? this should be clarified. The same applies to flexibility.
- Add discussion of the possible clinical implication of the findings (e.g., ergonomic or physical intervention recommendations).
-To deepen in why no relationship was found with lumbar proprioception.
-The conclusion is clear but could benefit from a sentence emphasizing the applied potential of the findings in clinical or occupational practice.
Reviewer 3 Report
Comments and Suggestions for Authors
I would like to appreciate your hard work in carrying out this research.
General concept comments:
- While multiple regression analysis was utilized, but the manuscript does not provide details of the methodology, including assumption checks and the corresponding standard results. Addressing this gap would enhance the rigor and transparency of the analysis.
- Several contributing factors to lumbar-pelvic rhythm were identified and measured. However, the rationale for their selection is not fully elaborated. Providing more information helps establish that the parameters are well-suited to address the research questions.
- It is not clear how the estimated regression coefficients were consulted to compare the related factors between the two groups, particularly given that there were parameters for which data were not recorded in the group without low back pain (e.g. VAS, ODI, FreBAQ). Clarifying this point would strengthen the understanding of the analysis and its implications.
Specific comments:
- Page 1, Lines 21-27: In the Method section of the Abstract, only measurements have been mentioned without further details.
- Page 1, Lines 30-31: The sentence fits better to the Results section of the Abstract.
- Page 2, Lines 54-56: It would be beneficial to mention other instances, where lumbar-pelvic rhythm was compared between individuals with and without low back pain. Also, it is not clear, why “movement” was indicated, while the focus of the study is on sedentary tasks.
- Page 4, Lines 128-129: it is better to present the equation as a Formula, not as a Figure.
- Page 5, Lines 167-173: Graphical representation of the calculated angles would be more informative.
Aim to establish a balance between active and passive voice throughout the manuscript.
Round 2
Reviewer 1 Report
Comments and Suggestions for Authors
According to the authors' supplementary literature (No. 34-36), Boonstra et al. (2014) pointed out that "VAS scores ≤34 mm were best described for patients with chronic musculoskeletal pain as mild pain." The VAS of the low back pain group in this study was 26.1 mm, which is a mild grade. Mazur et al. (2015) pointed out that ODI < 40% is classified as mild to moderate, while ODI ≥ 40% is classified as severe. The ODI of the low back pain group in this study was 15.7%, which is classified as mild. Mahmoudzadeh et al. (2020) did not mention the FreBAQ scores of the low back pain group. Therefore, these literature cannot support the authors' claim that "these values ​​indicate mild pain, functional impairment, and mild to moderate physical perception impairment". Furthermore, the authors have not provided the VAS, ODI, and FreBAQ scores of the Non-LBP group, so it is impossible to determine whether the degree of low back pain in this group of subjects meets the condition of “no low back pain”. The value of this study is affected by doubts about the subjects' qualifications.
Author Response
We sincerely appreciate your valuable feedback. Please refer to the attached file for our point-by-point responses and the corresponding revisions made in the manuscript.

Reviewer 2 Report
Comments and Suggestions for Authors
Most of the suggestions made by the reviewer have been addressed.
Author Response
Thank you for your constructive feedback. We have carefully revised the manuscript and addressed most of the suggestions provided. We believe the current version reflects these improvements, and we sincerely appreciate your guidance in enhancing the quality of our work.
Reviewer 3 Report
Comments and Suggestions for Authors
I appreciate the corrections and revisions made to the manuscript.
Please, consider the following comments that have not yet been addressed.
- The study follows a cross-sectional design, in which the outcome is treated as the dependent variable. Besides, multiple regression has been employed for the analyses. Nevertheless, there are only one dependent variable and several independent variables defined for this study (Page 7, Lines 274-281). This is neither consistent with the nature of a cross-sectional design, nor a multiple regression method. Whether this is a simple typing error or a methodological error, it should be corrected.
- Several contributing factors to lumbar-pelvic rhythm were identified and measured. However, the rationale for their selection is not fully elaborated. Providing more information helps establish that the parameters are well-suited to address the research questions.
- For better consistency, the graphical representation in Figure 2 could be redesigned to match the format of the other figures in the manuscript.
Author Response

(The authors gave the same response as above.)
